# Assessment of psychological terror and its impact on mental health and quality of life in medical residents at a reference medical center in Mexico: A cross-sectional study

Víctor Manuel Enriquez Estrada[1,2]* , Neftali Eduardo Antonio-Villa[3], Omar Yaxhemen Bello-Chavolla[4], Carlos Fredy Cuevas-García[1,2,3,4], Pedro Luis Vargas Gutiérrez[5], Irma Sau-Yen Corlay Noriega[1], Luis Rey García-Cortés[6]

1 UMAE, Hospital de Especialidades Centro Médico Nacional Siglo SXXI, Instituto Mexicano del Seguro Social, Ciudad de México, México, 2 Hospital General de Zona 197, Instituto Mexicano del Seguro Social, Ciudad de México, México, 3 Departamento de Endocrinología, Instituto Nacional de Cardiologia Ignacio Chávez, Ciudad de México, México, 4 Dirección de Investigación, Instituto Nacional de Geriatría, Ciudad de México, México, 5 Coordinación de Planeación y Enlace Institucional, Instituto Mexicano del Seguro Social, Ciudad de México, México, 6 Coordinador Auxiliar Medico de Investigacion en Salud, Instituto Mexicano del Seguro Social, Ciudad de México, México

☯ These authors contributed equally to this work.

* victor.enriquez.estrada.imss@gmail.com, vicmanuel280@gmail.com

## Abstract

### Background

Mobbing, particularly in medical residencies, can lead to psychological terror with lasting mental and physical health consequences. Its impact on Mexican residents, however, remains underexplored.

### Aim

This study aimed to investigate the prevalence and associated factors of psychological terror among medical residents at a medical center in Mexico City.

### Methods

In a cross-sectional study, medical residents from various specialties were assessed for mobbing domains, quality of life, and anxiety/depression using the Leymann Inventory of Psychological Terror (LIPT), 36-Item Short Form Health Survey, Beck Depression Inventory-II, and Beck Anxiety Inventory, respectively. Psychological terror was defined as a LIPT score ≥ p80. Linear and binomial logistic regression models were used to explore independent predictors of mobbing and psychological terror.

### Results

Of the 349 participants included (median age: 28; IQR: 27–30 years), 19.5% (95% CI: 15.5%-24.0%) were identified with psychological terror. Furthermore, 39% reported higher-degree trainees as mobbing perpetrators. Women in surgical residencies in their second or

**Data Availability Statement:** All relevant data are available on Github: https://github.com/neftalivilla/psychological_terror.

**Funding:** The author(s) received no specific funding for this work.

**Competing interests:** The authors have declared that no competing interests exist.

fifth year were found to experience higher levels of mobbing. Manifested bullying, workplace stigma, and inappropriate tasks were the most impacted mobbing domains. Anxiety, diminished mental health quality of life, and higher degree of medical specialization were independent predictors of mobbing. Meanwhile, increased anxiety, affiliation to surgical specialties, and being in the second or fifth year of training were identified as predictors of psychological terror.

## Conclusions

Mobbing and psychological terror are prevalent conditions among medical residents in Mexico. Identification of occupational conditions and adverse psychological stressors can help to improve quality of life and training of medical residents.

## Introduction

Mobbing in the workplace is a situation in which individuals experience harassment or intimidation from a coworker, often in hierarchical settings, thus leading to psychological terror [1]. Leyman H. introduced the term "psychological terror" to reflect a state of prolonged, continuous, and systematically perpetuated mobbing directed at specific individuals, pushing them into a helpless and defenseless position in the workplace [1, 2]. Consequently, psychological terror is a direct effect of workplace mobbing. It's been documented that mobbing and psychological terror negatively impact both the mental and physical health of workers, leading them towards negative lifestyle habits and psychological distress [3]. Furthermore, mobbing can promote anxiety, depression, sleep disturbances, apathy, diminished self-esteem, and even encourage suicidal behaviors [4, 5].

Although anyone can experience mobbing, healthcare workers, particularly those in medical residencies, have been identified as a vulnerable group susceptible to workplace harassment and, consequently, varying degrees of mobbing [6, 7]. However, the evidence regarding the assessment of mobbing or psychological terror in medical workplaces is limited and typically focuses solely on workplace harassment. A meta-analysis comprising data from 11 countries reported that 63.4% of all physicians in medical residencies experienced some degree of workplace harassment, with gender discrimination being the most common type [6]. Moreover, studies have shown that the prevalence of workplace harassment in medical residencies in Latin America is higher compared to other regions globally [8–10]. Research conducted in northern Mexico found a 93% prevalence of workplace harassment in a pediatric hospital, with physical (32%) and psychological (15%) violence being the most common types, and 20% of these incidents involving extreme violence [11]. In addition, a recent study indicated that 98.5% of medical participants had experienced at least one mobbing behavior in the past six months, with gynecology and obstetrics specializations reporting the highest psychological terror metrics [11–13].

Despite having over 160,000 in-training medical residents across Mexico, there are numerous opportunities to improve mental health, quality of life, and working conditions in this population. Furthermore, efforts are needed to quantify the extent of mobbing and psychological terror across medical residencies, which could result in improved working conditions and educational policies within medical training programs. In this study, we aimed to 1) evaluate mobbing domains and the prevalence of psychological terror and 2) assess the associated

factors for mobbing and psychological terror within a sample of medical residents enrolled in a cross-sectional study at a reference medical center in Mexico City.

## Methods

### Study design

We conducted an observational cross-sectional study. Our sample consisted of medical residents from the XXI[st] Century National Medical Center, a third-level medical center in Mexico City. The study period during which we collected our data spanned from January 2019 to April 2019. We invited medical residents to voluntarily answer a questionnaire, which included socio-demographic variables (age, sex, medical residency, year and stage of residency), and four psychometric tests: the Leymann Inventory of Psychological Terror (LIPT), the 36-Item Short Form Health Survey (SF-36), the second version of the Beck Depression Inventory (BDI-II), and the Beck Anxiety Inventory (BAI-II) [14–18]. In Mexico, there are 27 medical residencies available for direct application [19]. For convenience, we grouped these into four main types of medical residencies: 1) Internal Medicine-related areas, 2) Surgical specialties, 3) Diagnostic residencies such as Radiology, Audiology (offered as part of physician training programs), and Pathology, and 4) Psychiatry. All participants provided electronic informed consent, and all gathered information was anonymized. Approval from the Ethics Committee was waived given that the information presented was extracted from an open call for volunteers to participate in this study. This study adhered to the principles of the Declaration of Helsinki and the STROBE guidelines for cross-sectional studies (S1 Table).

### Outcome variables definition

The main analyses focused on two primary outcomes: mobbing and psychological terror.

a) Mobbing—We used the LIPT test, a tool designed to measure mobbing in the workplace. The LIPT test aims to capture the frequency and intensity of negative conduct in the workplace that could potentially lead to psychological terror. Higher LIPT scores indicate greater perceived mobbing. For this study, we used a Spanish version of the LIPT that has been previously validated in the Mexican population [18]. To better characterize mobbing, we explored the six main domains of the LIPT test: laboral and personal stigma, overt and covert bullying, inappropriate communication, and inappropriate tasks assigned at work. Examples of its application and implications for healthcare workers have been published elsewhere [6].

b) Psychological terror–Given that psychological terror represents the most severe degree of experienced mobbing, we defined it as subjects with an overall LIPT score above the 80[th] percentile (p≥80) as described within the LIPT inventory [1].

### Quality of life assessment

We employed the SF-36 instrument, a questionnaire previously validated for the Mexican population, to measure self-perceived quality of life [17]. The SF-36 contains items formulated as questions to identify eight specific health constructs. Higher SF-36 scores indicate a more positive health perception. Our analysis sought to describe the two main global domains of quality of life: physical health (PCS) and mental health (MCS).

## Measurement of depression and anxiety

Depression was assessed using the second version of the BDI-II. We employed a Spanish version of the BDI-II, standardized for the Mexican population, where a higher score suggests more severe depression [15]. Anxiety was evaluated using the second version of the BAI-II, also in its Spanish version standardized for the Mexican population [14]. Similarly, higher BAI scores indicate elevated levels of anxiety. Both depression and anxiety were categorized into four levels: minimal, mild, moderate, and severe, according to previously validated thresholds.

## Statistical analysis

Categorical variables were represented using frequencies and their corresponding percentages. Continuous variables were described using either the mean (standard deviation) or median (inter-quartile range [IQR]), depending on their underlying distribution. To compare the descriptive characteristics of subjects with psychological terror, we employed either a Student's t-test or Mann-Whitney test for continuous variables, as appropriate, and Pearson's chi-squared test for categorical variables.

## Prevalence of psychological terror

To estimate the prevalence of psychological terror, we used the Clopper-Pearson method from the *epiR* package (Version 2.0.3) [20]. To further evaluate the prevalence of psychological terror, we stratified it by the four main types of specialties in medical residencies and by year of residency; both were further stratified by sex.

## Independent factors related to mobbing and psychological terror

We fitted Spearman correlations and linear regression models to evaluate predictors of LIPT values, using quality of life, depression, and anxiety levels as the main associated predictors of mobbing, adjusting for age, sex, and type of medical residencies. The same variables were then tested as predictors of LIPT p≥80 using a logistic regression model to evaluate the phenotype of overt psychological terror. Regression diagnostics were performed using residual analyses for the linear models, and the Hosmer-Lemeshow test for the logistic model. All statistical analyses were performed using R version 4.1.2 [21].

# Results

## Characteristics of the studied population

Our study sample comprised 349 medical residents undergoing specialization training during our study period. Complete sociodemographic and psychometric characteristics are presented in Table 1. Briefly, we observed a slight male predominance (50.4%) with a median age of 28 years (IQR 27–30). The distribution of medical specializations was as follows: Internal Medicine (41.8%), Surgery (24.9%), Diagnostics (pathology, audiology, and radiology) (20.9%), and Psychiatry (12.3%). Most of our participants, 77.3%, were residents in their 1st (24.9%), 2nd (27.5%), and 3rd (24.9%) year of medical residency. The median LIPT score in our sample was 39.2 (IQR 36.2–45.6) points.

## Characterization of psychological terror

We identified that 19.5% (95% CI: 15.5%-24.0%) of our study population exhibited psychological terror or scored above the 80th percentile of LIPT (>23 points). In subjects with psychological terror (n = 68), we observed a higher number of residents from surgical specialties

**Table 1. Sociodemographic, medical, and psychometric scales variables from the studied population and stratified between subjects without physiological terror.**

| Parameter | All-Subjects (n = 349) | Without Physiological Terror (n = 281) | Physiological Terror (n = 68) | P-Value | Statistic Value |
|---|---|---|---|---|---|
| Age (years) | 28 (27–30) | 28 (27–30) | 28 (27–30) | 0.568 | W = 4585.5 |
| Men (%) | 176 (50.4) | 144 (51.25) | 32 (47.06) | 0.6281 | $\chi^2 = 0.234$ |
| *Medical Specialties* | | | | | |
| Internal Medicine (%) | 146 (41.8) | 123 (43.77) | 23 (33.82) | 0.152 | $\chi^2 = 1.836$ |
| Surgery (%) | 87 (24.9) | 57 (20.28) | 30 (44.12) | <0.001 | $\chi^2 = 15.369$ |
| Radiology, Audiology and Pathology (%) | 73 (20.9) | 64 (22.78) | 9 (13.24) | 0.107 | $\chi^2 = 2.463$ |
| Psychiatry (%) | 43 (12.3) | 37 (13.17) | 6 (8.82) | 0.429 | $\chi^2 = 0.596$ |
| 1st (%) | 87 (24.9) | 75 (26.69) | 12 (17.65) | 0.164 | $\chi^2 = 1.933$ |
| 2nd (%) | 96 (27.5) | 71 (25.27) | 25 (36.76) | 0.079 | $\chi^2 = 3.076$ |
| 3rd (%) | 87 (24.9) | 73 (25.98) | 14 (20.59) | 0.443 | $\chi^2 = 0.586$ |
| 4th (%) | 59 (16.9) | 50 (17.79) | 9 (13.24) | 0.471 | $\chi^2 = 0.517$ |
| 5th (%) | 20 (5.7) | 12 (4.27) | 8 (11.76) | 0.036 | $\chi^2 = 4.389$ |
| *Psychometric Scales* | | | | | |
| Physical Quality of Life (Pts) | 39 (36–45) | 38 (36–43) | 44 (37–50) | <0.001 | W = 6435 |
| Mental Quality of Life (Pts) | 45 (39–50) | 46 (40–50) | 40 (36–46) | <0.001 | W = 11,937 |
| BAI-Anxiety Levels (Pts) | 5 (1–13) | 4 (1–10) | 13 (5–21) | <0.001 | W = 5427.5 |
| Minimal-Anxiety (%) | 177 (50.7) | 159 (56.58) | 18 (26.47) | <0.001 | $\chi^2 = 18.677$ |
| Mild-Anxiety (%) | 109 (31.2) | 86 (30.6) | 23 (33.82) | 0.651 | $\chi^2 = 0.13548$ |
| Moderate-Anxiety (%) | 49 (14) | 34 (12.1) | 15 (22.06) | 0.042 | $\chi^2 = 3.7122$ |
| Severe-Anxiety (%) | 14 (4) | 2 (0.71) | 12 (17.65) | <0.001 | $\chi^2 = 36.501$ |
| BDI-Depression Levels (Pts) | 4 (0–10) | 3 (0–8) | 10 (4–18.25) | <0.001 | W = 5475.5 |
| Minimal-Depression (%) | 261 (74.8) | 228 (81.14) | 33 (48.53) | <0.001 | $\chi^2 = 29.17$ |
| Mild-Depression (%) | 50 (14.3) | 35 (12.46) | 15 (22.06) | 0.06645 | $\chi^2 = 3.3686$ |
| Moderate-Depression (%) | 33 (9.5) | 14 (4.98) | 19 (27.94) | <0.001 | $\chi^2 = 31.081$ |
| Severe-Depression (%) | 5 (1.4) | 4 (1.42) | 1 (1.47) | 0.891 | $\chi^2 = 0.0001$ |
| LIPT-60 (pts) | 39.2 (36.2–45.6) | 38.8 (36–43.8) | 44.9 (37.5–50.5) | <0.001 | W = 7129.0 |

*Abbreviations*: BAI = Beck Anxiety Inventory; BDI = Beck Depression Inventory; $\chi^2$ = Chi-Squared statistic; W = Wilcoxon rank sum statistic.

(44.1%), mainly in their second year of training (11.7%), exhibiting decreased levels of physical and mental quality of life along with increased levels of anxiety and depression. There was a higher proportion of severe anxiety (17.6%) and moderate depression (27.9%) compared to subjects without psychological terror. The prevalence of psychological terror varied across types of medical residencies, with surgical residencies having the highest prevalence at 34.5% (95% CI: 24.6%-45.4%), followed by internal medicine at 15.8% (95% CI: 10.3%-22.7%). Psychiatry and the combined group of radiology, audiology, and pathology residents had prevalences of 13.9% (95% CI: 5.3%-27.9%) and 12.3% (95% CI: 5.8%-22.1%) respectively. Medical residents in their 2nd (26%, 95% CI: 17.6%-36.0%) and 5th (40%, 95% CI: 19.1%-63.9%) years experienced the highest prevalence of psychological terror, and this trend was even higher for women (Fig 1).

## Affected domain of mobbing

On examining the domains of the LIPT scale among subjects with psychological terror in our sample, manifested bullying, laboral stigma, and inappropriate tasks were the most affected

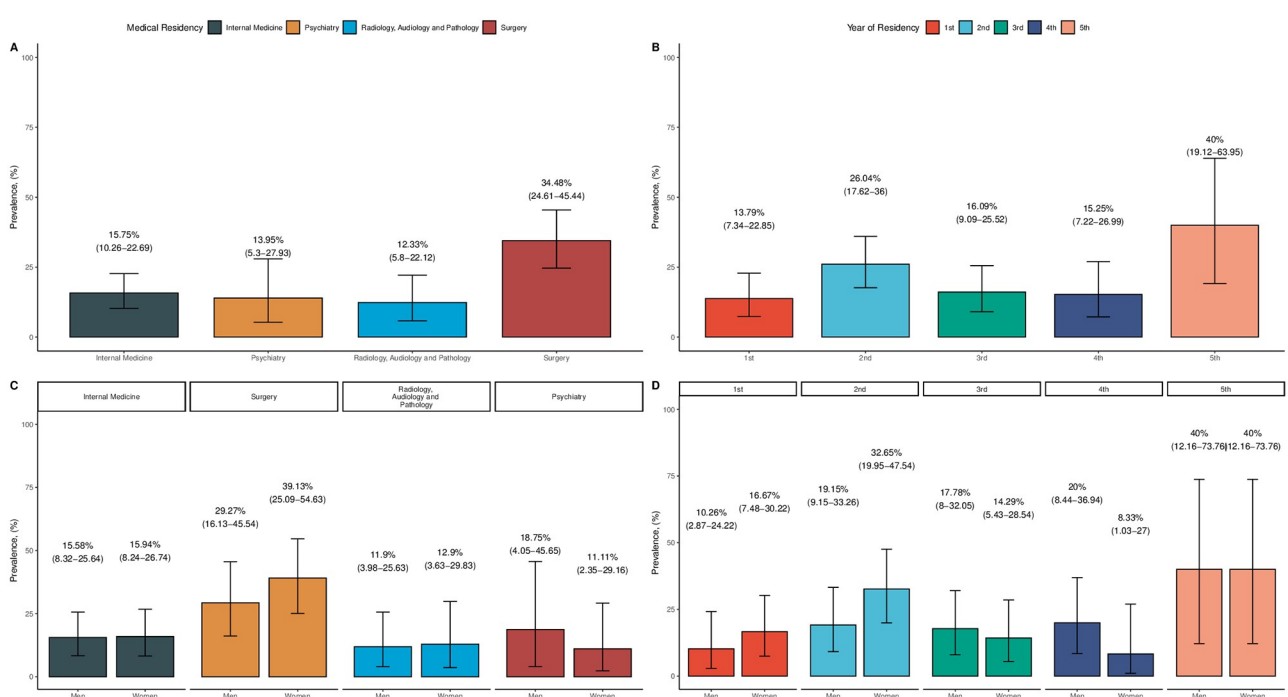

**Fig 1. Prevalence of psychological terror across medical residencies (A) and medical years (B) stratified by sex (C, D).**

domains. Stratifying by medical residencies, we observed that Internal Medicine and Surgical residencies tended to have higher rates of manifested bullying and laboral stigma. Diagnostic residencies reported higher instances of uncovered bullying and inappropriate tasks, while Psychiatry residents reported a higher personal stigma among peers. Stratifying the domains according to residency years, we observed that residents in their 3rd year experienced the highest laboral stigma, manifested, and uncovered bullying scores (Fig 2). We asked participants to identify individuals who perpetrated harassing conduct most frequently and found that peers at higher hierarchical positions showed the highest frequency (39%), followed by heads of medical staff (21.2%) (Table 2).

## Independent parameters associated with mobbing

First, we assessed the partial correlations among the LIPT inventory scores to determine the factors associated with mobbing in our population, adjusting for age, sex, type of residency, and level of education. LIPT scores showed a positive correlation with BAI-II ($\rho$ = 0.444, 95% CI: 0.297–0.575), BDI-II ($\rho$ = 0.359, 95% CI: 0.232–0.478), and SF-36-PCS ($\rho$ = 0.152, 95% CI: 0.034–0.263), and a negative correlation with SF-36-MCS ($\rho$ = -0.315, 95% CI: -0.425 to -0.164). Next, to identify independent factors associated with higher LIPT-60 scores, we fitted a multiple linear regression model. Higher LIPT-60 scores were associated with higher BAI-II scores ($\beta$ = 0.948, p<0.001), lower SF-36-MCS ($\beta$ = -0.404, p = 0.025), and more advanced years of residency ($\beta$ = 2.549, p = 0.009). These associated factors explained 19.1% of the variability in the LIPT scale. Consequently, we could consider increased years of residency and anxiety levels as well as decreased mental quality of life as the main determinants of mobbing (Table 3).

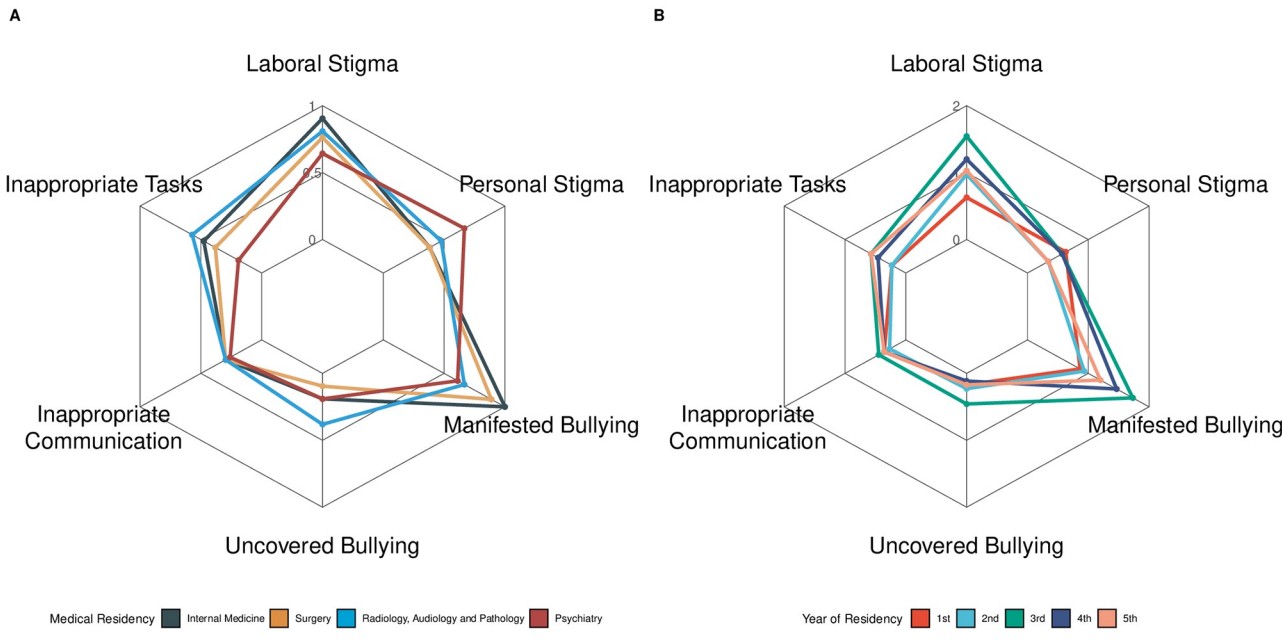

**Fig 2. Median score of each domain of LIPT across medical residencies (A) and medical years (B).**

## Determinants of psychological terror

We used logistic regression to investigate factors associated with LIPT p≥80. Subjects with psychological terror had higher BAI-II scores (OR 1.10, 95% CI: 1.06–1.13), were more likely to belong to surgical specialties (OR 2.57, 95% CI: 1.27–5.21), and were more likely to be in their second (OR 2.73, 95% CI: 1.15–6.50) or fifth year (OR 5.54, 95% CI: 1.59–19.37) of medical residency. The determinants identified in the model explained 26.7% of the variability of the highest LIPT p≥80 among the surveyed medical specialties residents. Therefore, we could consider that having anxiety, belonging to surgical specialties, and being in the second or fifth year of training were the main determinants of psychological terror (Table 3).

**Table 2. Frequency and percentage of those responsible for the open question: "Who has mobbing or harassed you at work?".**

| Who has mobbing or harassed you at work? | n = 349 |
|---|---|
| Senior classmates | 136 (39.0%) |
| Heads of medical staff | 74 (21.2%) |
| No-one | 52 (14.9%) |
| Other medical staff | 38 (10.9%) |
| Nursing | 16 (4.6%) |
| Peers in the same degrees | 22 (6.3%) |
| Peers from lower degrees | 4 (1.1%) |
| Attached chiefs | 5 (1.4%) |
| Patient and relatives | 1 (0.3%) |
| Everyone | 1 (0.3%) |

**Table 3. Linear regression and logistic regression models to identify factors associated with psychological terror.** Linear regression model adjusted for: age, sex, type of residence and degree of residence. Logistic regression model for LIPT-60 >p80 adjusted for: age and sex. Annotations: *Compared to Internal Medicine. **Compared to a first-year resident.

| Linear Regression | Parameter | Beta Coefficient | T-Value | P Value | IC95% |
|---|---|---|---|---|---|
| $r^2 = 0.199$<br>$F = 29.876$<br>$P<0.001$ | Intercept | 18.904 | 2.071 | 0.039 | 0.054–36.854 |
| | BAI | 0.948 | 6.611 | <0.001 | 0.666–1.231 |
| | SF-36 MCS | -0.404 | -2.253 | 0.025 | -0.757 to -0.051 |
| | Year or Residency | 2.549 | 2.618 | 0.009 | 0.634–4.463 |
| **Logistic Regression** | **Parameter** | **Beta Coefficient** | **Wald Test** | **P Value** | **OR (IC95%)** |
| $r^2 = 0.267$<br>Hosmer-Lemeshow = 7.848<br>$P = 0.448$ | Intercept | -3.362 | 43.692 | <0.001 | — |
| | BAI | 0.092 | 34.771 | <0.001 | 1.096 (1.063–1.130) |
| | Surgical Specialties* | 0.946 | 6.906 | 0.009 | 2.575 (1.272–5.214) |
| | Second year resident** | 1.004 | 5.148 | 0.023 | 2.729 (1.146–6.496) |
| | Fifth year resident** | 1.713 | 7.199 | 0.007 | 5.544 (1.587–19.37) |

## Discussion

In our study, we found a high prevalence of psychological terror across medical residencies in a cross-sectional sample from a third-level medical center in Mexico City. We discovered that residents in surgical specialties and those in their second or fifth year are more likely to experience mobbing, a trend more pronounced in women. Interestingly, our results suggest that the most common perpetrators of mobbing are medical residents in superior hierarchical positions, followed by medical staff. Additionally, we identified increased anxiety levels and decreased mental health-related quality of life as significant correlates of higher mobbing levels. Our study's findings enhance our understanding of the psychological terror phenomenon among medical residents in Mexico, potentially informing future strategies to mitigate this issue among Mexican and Latin American residents.

### Mobbing and psychological terror in medical residencies

Our findings align with several authors suggesting that medical trainees are at an elevated risk of experiencing psychological terror [9, 11, 12, 22]. While limited information exists on mobbing and psychological terror, several studies have attempted to describe this phenomenon as workplace harassment. One study conducted in a pediatric hospital in northern Mexico reported an alarmingly high prevalence of workplace harassment at 93%, with physical and psychological violence being the most common types (32% and 15%, respectively) [11]. Notably, 20% of violent harassment cases were classified as extreme violence. Another study using the LIPT score found that 256 (98.5%) of medical participants experienced at least one type of mobbing conduct within the preceding six months. In that study, residents in Gynecology and Obstetrics specializations reported the highest scores across all measures of psychological terror [12]. A different Mexican study involving 82 medical residents found that 18.3% had experienced severe physical violence [23]. Interestingly, the main perpetrators of workplace harassment in these studies were peers in higher-degree positions or medical staff personnel, consistent with our findings. Overall, our findings—while supported by previous research—underscore that mobbing is a prevalent condition among Mexican residents, ultimately leading to a significant burden of psychological terror.

### Factors influencing mobbing and psychological terror

Our study delineates specific factors influencing mobbing levels and the experience of psychological terror, which could be potentially targeted for interventions within medical residencies.

First, our results highlight a correlation between a higher degree of medical residency and higher levels of mobbing. Contrary to several studies suggesting lower hierarchical positions as a risk factor, we found increased mobbing among medical residents in their second and fifth years [6, 8, 11, 22, 24]. This discrepancy may be attributed to an accumulation of psychological erosion and burnout syndrome specific to our study population. A higher position in the hospital hierarchy involves supervisory duties, greater administrative tasks, and the performance of more complex procedures, all of which may expose these individuals to greater punishment and harassment from superiors. Prior studies have made similar observations, linking culpability, shame, and reduced satisfaction from studying a medical specialization with mobbing [22, 24]. Overall, mobbing significantly impairs quality of life and adversely impacts both mental and physical health, leading to detrimental lifestyle habits and psychological distress [3].

## Impact of mobbing and psychological terror in medical residencies

The impact of psychological terror on physicians' quality of life remains underexplored in the literature [25]. Our study reveals a correlation between psychological terror among medical residents and reduced mental component scores on the SF-36 quality-of-life scale, a factor previously unreported in healthcare workplace harassment studies. This connection is indicative of a significant, potentially preventable, effect. The current medical education model in our country prioritizes the assessment of medical knowledge and executive tasks, often overlooking psychometric and psychological evaluations throughout residency programs. Consequently, medical residents experience heightened levels of anxiety, depression, and poor quality of life. Certain residents may replicate and perpetuate negative coping strategies, such as passivity towards verbal aggression, making them more susceptible to mobbing, workplace erosion, and thus, reduced professional performance and ultimately, psychological terror [26, 27]. The third novel finding from our study is the characterization of psychological terror manifestations by medical specialty. Our results highlight a significant prevalence of psychological terror, particularly manifest violence, among surgical residents. This finding starkly contrasts with specialties like psychiatry, where violence is primarily directed towards individual disrepute. We observed diverse patterns of mobbing, potentially attributable to varying resident personality profiles, specialty choices, and sociodemographic determinants relating to working conditions in Mexico. Our findings underscore the need for health authorities to implement interventions aimed at detecting mobbing and minimizing its impact on Mexican medical residents.

## Strengths and limitations

Our study boasts several strengths and limitations. Its main strength lies in its size, being the most extensive study of mobbing and psychological terror among Mexican medical residents, encompassing multiple medical specialties at a leading tertiary hospital. Our findings provide a snapshot of the workplace environment for medical residents in Mexico today. To mitigate volunteer bias, we ensured participants understood the study's objective, the anonymity of their responses, and the voluntary nature of their involvement, emphasizing that participation would not impact their professional status. Furthermore, we could propose novel independent predictors of increased psychological terror using statistical models, which could be investigated in future studies. Regarding limitations, as a cross-sectional study, we cannot infer causality between psychological terror and the other evaluated parameters. We did not consider potentially influential variables, such as personality traits, substance abuse, level of work erosion, and sociodemographic conditions, which may also affect susceptibility to and vulnerability to workplace mobbing. Consequently, we recommend further research through

longitudinal studies that can evaluate these parameters, already associated with reduced quality of life, depression, and anxiety. Lastly, given our study's reliance on self-reported questionnaires, we must acknowledge a potential selection bias towards participants who volunteered for our study, who may not represent the entire population of Mexican medical residents.

## Conclusions

In our study, we found that nearly one in every five medical residents exhibited psychological terror. Manifest bullying, workplace stigma, and inappropriate tasks emerged as the most impaired domains of mobbing. Women in surgical residencies in their second or fifth year were found to experience higher levels of mobbing. The perpetrators of mobbing were most frequently higher-level medical residents and medical staff. Psychological terror was found to be associated with female sex, higher anxiety levels, lower mental quality of life, and higher hierarchical positions during training. It is crucial to identify and confront mobbing and psychological terror in the workplace, along with their associated independent factors. Doing so represents could be a significant stride towards minimizing and controlling the psychological harm resulting in an improvement in quality of life and training of medical residents.

## Supporting information

**S1 Table. STROBE cohort guidelines report for study.** This checklist was completed on 22. May 2023 using https://www.goodreports.org/ a tool made by the EQUATOR Network in collaboration with Penelope.ai (von Elm E, Altman DG, Egger M, Pocock SJ, Gotzsche PC, Vandenbroucke JP. The Strengthening the Reporting of Observational Studies in Epidemiology (STROBE) Statement: guidelines for reporting observational studies).
(DOCX)

**S1 File.**
(DOCX)

## Author Contributions

**Conceptualization:** Víctor Manuel Enriquez Estrada, Neftali Eduardo Antonio-Villa, Carlos Fredy Cuevas-García.

**Data curation:** Omar Yaxhemen Bello-Chavolla, Carlos Fredy Cuevas-García.

**Formal analysis:** Neftali Eduardo Antonio-Villa, Omar Yaxhemen Bello-Chavolla.

**Funding acquisition:** Víctor Manuel Enriquez Estrada.

**Investigation:** Víctor Manuel Enriquez Estrada, Neftali Eduardo Antonio-Villa, Irma Sau-Yen Corlay Noriega, Luis Rey García-Cortés.

**Methodology:** Víctor Manuel Enriquez Estrada, Neftali Eduardo Antonio-Villa, Omar Yaxhemen Bello-Chavolla, Pedro Luis Vargas Gutiérrez, Irma Sau-Yen Corlay Noriega, Luis Rey García-Cortés.

**Project administration:** Víctor Manuel Enriquez Estrada, Carlos Fredy Cuevas-García, Pedro Luis Vargas Gutiérrez, Luis Rey García-Cortés.

**Resources:** Víctor Manuel Enriquez Estrada, Carlos Fredy Cuevas-García, Pedro Luis Vargas Gutiérrez.

**Supervision:** Víctor Manuel Enriquez Estrada, Luis Rey García-Cortés.

**Visualization:** Neftali Eduardo Antonio-Villa.

**Writing – original draft:** Neftali Eduardo Antonio-Villa.

**Writing – review & editing:** Víctor Manuel Enriquez Estrada, Neftali Eduardo Antonio-Villa, Omar Yaxhemen Bello-Chavolla.

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
