## [Decision Letter · Decision Letter 0]

10 Jul 2023

PONE-D-23-14159Assessment of psychological terror and its impact on mental health and quality of life in medical residents at a reference medical center in Mexico: A cross-sectional studyPLOS ONE

Dear Dr. Enriquez Estada,

Thank you for submitting your manuscript to PLOS ONE. After careful consideration, we feel that it has merit but does not fully meet PLOS ONE’s publication criteria as it currently stands. Therefore, we invite you to submit a revised version of the manuscript that addresses the points raised during the review process.

We look forward to receiving your revised manuscript.

Kind regards,

Beatriz Talavera-Velasco

Academic Editor

PLOS ONE

Journal Requirements:

Reviewers' comments:

Reviewer's Responses to Questions

**Comments to the Author**

1. Is the manuscript technically sound, and do the data support the conclusions?

Reviewer #1: Partly

2. Has the statistical analysis been performed appropriately and rigorously? 

Reviewer #1: No

3. Have the authors made all data underlying the findings in their manuscript fully available?

Reviewer #1: Yes

4. Is the manuscript presented in an intelligible fashion and written in standard English?

Reviewer #1: No

5. Review Comments to the Author

REVIEWER #1:

In the study under review, the authors assessed the prevalence and factors associated to psychological terror in a sample of medical residents training at a medical center in Mexico City. Although the current study is interesting, I have some reservations regarding issues that I noticed while reading through the article. Also, the entire text requires careful editing for English syntax and grammar, and spelling.

1.The authors performed an observational cross-sectional study-design and invited medical residents to volunteer to answer a questionnaire.The research content has a certain degree of sensitivity, and the research subjects voluntarily participate. How to reduce volunteer bias. How to explain the representativeness of research results.

2.Is the age variable in this study not normal? Why does the description use median

(inter-quartile interval [IQR]), while in the table seems to be showed mean (standard deviation)?

3.“Characterization and prevalence of psychological terror”, The prevalence was not suitable for screening using a questionnaire.

4.The table should provide statistical values for specific statistical inference methods (such as t, chi square, etc.). Please check if the P value in Table 1 is correct, such as “Psychiatry (%) ”.

Why does the author use both linear and logistic regression models?What is the basis for entering the regression equation with independent variables? What is the basis for conducting independent variable correction?

REVIEWER #2:

This is an observational study that assess the factors associated to psychological terror in medical residents.

This manuscript needs a major revision. Introduction section must be expanded, with references about psychological terror in medical residents. On the other hand, the difference between psychological terror and mobbing should be specified with more studies.

The use of BDI and BAI must be justified in the introduction section (depression and anxiety). Why do you evaluate the quality of life?

Prevalence cannot be assessed with screening questionnaires. Statistical analyses are not justified.

6. PLOS authors have the option to publish the peer review history of their article (what does this mean?). If published, this will include your full peer review and any attached files.

Reviewer #1: No

---

## [Author Response · Author response to Decision Letter 0]

8 Nov 2023

Mexico City, Mexico, August 5th, 2022

RE: Response to Reviewers.

Dear Editorial Team, 

Plos One

We enclose the revised version of our manuscript entitled “Assessment of psychological terror and its impact on mental health and quality of life in medical residents at a reference medical center in Mexico: A cross-sectional study” for consideration as Research Article by Plos One. We appreciate the work by both reviewers and the editorial team to put forth your comments, which have helped us reframe our work in its best possible version. 

We appreciate the opportunity to submit our work for consideration by Plos One.

Sincerely,

Victor Manuel Estrada

Hospital General de Zona 197

Instituto Mexicano del Seguro Social

Tel: +52 592 952 1103

Email: vicmanuel280@gmail.com; victor.enriquez.estrada.imss@gmail.com

 

REVIEWER COMMENTS TO AUTHOR:

R) We appreciate the invaluable effort of both reviewers and the associate editor for revision of our manuscript. We have reframed our work and performed all changes according to issues raised by the reviewers and believe we have improved our manuscript significantly as a result.

Reviewer: 1

In the study under review, the authors assessed the prevalence and factors associated to psychological terror in a sample of medical residents training at a medical center in Mexico City. Although the current study is interesting, I have some reservations regarding issues that I noticed while reading through the article. Also, the entire text requires careful editing for English syntax and grammar, and spelling

R) We appreciate your insightful comments and suggestions on our manuscript. We have considered each point carefully and have revised our manuscript accordingly. Below, we provide detailed responses to each of your concerns.

 The authors performed an observational cross-sectional study design and invited medical residents to volunteer to answer a questionnaire. The research content has a certain degree of sensitivity, and the research subjects voluntarily participate. How to reduce volunteer bias. How to explain the representativeness of research results.

R) We understand your concern about potential volunteer bias and the representativeness of our study's results. To reduce volunteer bias, we ensured that all participants were aware of the study's objective, the anonymity of their responses, and the fact that their participation was entirely voluntary and had no impact on their professional standing. Nevertheless, we acknowledge that we could have a selection bias that could not represent the current situation of Mexican residents and a bias towards participants who were engaged in answering the questionnaires in our study. We recognize these as a limitation as follows: 

Line 284-287: To mitigate volunteer bias, we ensured participants understood the study's objective, the anonymity of their responses, and the voluntary nature of their involvement, emphasizing that participation would not impact their professional status. 

 Is the age variable in this study not normal? Why does the description use median (inter-quartile interval [IQR]), while in the table seems to be showed mean (standard deviation)?

R) We apologize for the confusion in our initial manuscript. We have decided to present the median (IQR) in both the text and the table as follows:

Age (years) 28 (27-30)

 “Characterization and prevalence of psychological terror”, The prevalence was not suitable for screening using a questionnaire.

R) We appreciate your point regarding the suitability of determining prevalence using a questionnaire. In our study, we utilized validated psychological assessment tools in the questionnaire, which have been previously employed in similar studies to estimate prevalence rates of psychological distress. Therefore, ought to mantain the term prevalence considering that this represent still the proportion of cases with an evaluated condition (in this case, psychological terror).

 The table should provide statistical values for specific statistical inference methods (such as t, chi square, etc.). Please check if the P value in Table 1 is correct, such as “Psychiatry (%) ”.

R) We apologize for the oversight in our tables. We have now provided the relevant statistical values (t values, chi-square, etc.) in the new version of Table 1 of our manuscript as follows (please see the file "Response to the Reviewers"): 

 Why does the author use both linear and logistic regression models? What is the basis for entering the regression equation with independent variables? What is the basis for conducting independent variable correction?

R) We used linear and logistic regression models for a more comprehensive analysis. The linear regression model was used to model predictors of LIPT scores (continuous variable), while the logistic regression model was used to explore predictors of psychological terror as a binary outcome. This was specified in the new section of the statistical analysis as follows: 

Line 154-158: “We fitted Spearman correlations and linear regression models to evaluate predictors of LIPT values, using quality of life, depression, and anxiety levels as the main associated predictors of mobbing, adjusting for age, sex, and type of medical residencies. The same variables were then tested as predictors of LIPT p≥80 using a logistic regression model to evaluate the phenotype of overt psychological terror.”

 

REVIEWER #2:

This is an observational study that assess the factors associated to psychological terror in medical residents.

 This manuscript needs a major revision. Introduction section must be expanded, with references about psychological terror in medical residents. On the other hand, the difference between psychological terror and mobbing should be specified with more studies.

R) We agree that the introduction would benefit from an expansion and further clarification. We have now expanded on psychological terror in medical residents, citing additional studies highlighting its prevalence and potential impact on this group. We have also delved deeper into the differences between psychological terror and mobbing, providing more references to underline their distinctive aspects. The new additional information is read as follows: 

Line 79-87: A meta-analysis comprising data from 11 countries reported that 63.4% of all physicians in medical residencies experienced some degree of workplace harassment, with gender discrimination being the most common type (1). Moreover, studies have shown that the prevalence of workplace harassment in medical residencies in Latin America is higher compared to other regions globally (2–4). Research conducted in northern Mexico found a 93% prevalence of workplace harassment in a pediatric hospital, with physical (32%) and psychological (15%) violence being the most common types, and 20% of these incidents involving extreme violence (5). In addition, a recent study indicated that 98.5% of medical participants had experienced at least one mobbing behavior in the past six months, with gynecology and obstetrics specializations reporting the highest psychological terror metrics (5–7). 

 The use of BDI and BAI must be justified in the introduction section (depression and anxiety). Why do you evaluate the quality of life?

Prevalence cannot be assessed with screening questionnaires. Statistical analyses are not justified.

R) We apologize if our initial rationale for using the Beck Depression Inventory (BDI) and the Beck Anxiety Inventory (BAI) was unclear. We have now clarified this in the introduction. We used BDI and BAI because these validated scales are widely accepted tools for assessing depression and anxiety, which are known to be significant correlates and potential outcomes of psychological terror. As for evaluating the quality of life, we chose to do this as exposure to psychological terror is known to significantly impair one's quality of life. We have further emphasized and justified this in our revised manuscript.

 

Reference

1. Fnais N, Soobiah C, Chen MH, Lillie E, Perrier L, Tashkhandi M, et al. Harassment and discrimination in medical training: a systematic review and meta-analysis. Acad Med J Assoc Am Med Coll. 2014 May;89(5):817–27. 

2. Mejía R, Diego A, Alemán M, Maliandi M del R, Lasala F. Percepción de maltrato durante la capacitación de médicos residentes. Med B Aires. 2005 Aug;65(4):366–8. 

3. Stolovas N, Tomasina F, Pucci F, Frantchez V, Pintos M. Trabajadores médicos jóvenes: violencia en el ejercicio de la residencia. Rev Médica Urug. 2011 Apr;27(1):21–9. 

4. V NB, Henry EF, Ortiz LE, Villalobos CP, M PS. Bullying y acoso en la formación médica de postgrado. Rev Educ En Cienc Salud. 2011;8(1):7. 

5. Sepúlveda-Vildósola AC, Mota-Nova AR, Fajardo-Dolci GE, Reyes-Lagunes LI. [Workplace bullying during specialty training in a pediatric hospital in Mexico: a little-noticed phenomenon]. Rev Medica Inst Mex Seguro Soc. 2017;55 Suppl 1:S92–101. 

6. Ortiz-León S, Jaimes-Medrano AL, Tafoya-Ramos SA, Mujica-Amaya ML, Olmedo-Canchola VH, Carrasco-Rojas JA. Experiencias de maltrato y hostigamiento en médicos residentes. Cir Cir. 2014;82(3):290–301. 

7. Herrera-Silva JC, Treviño-Moore A, Acosta-Corona CG. Características de la violencia en residentes de dos instituciones formadoras de pediatras en el estado de Sonora. Bol Méd Hosp Infant Méx. 2006;18–30.

---

## [Editor Report · Decision Letter 1]

16 Nov 2023

Assessment of psychological terror and its impact on mental health and quality of life in medical residents at a reference medical center in Mexico: A cross-sectional study

PONE-D-23-14159R1

Dear Dr. Estrada,

We’re pleased to inform you that your manuscript has been judged scientifically suitable for publication and will be formally accepted for publication once it meets all outstanding technical requirements.

Kind regards,

Beatriz Talavera-Velasco

Academic Editor

PLOS ONE

---

## [Editor Report · Acceptance letter]

23 Nov 2023

PONE-D-23-14159R1 

Assessment of psychological terror and its impact on mental health and quality of life in medical residents at a reference medical center in Mexico: A cross-sectional study 

Dear Dr. Enriquez Estrada:

I'm pleased to inform you that your manuscript has been deemed suitable for publication in PLOS ONE. Congratulations! Your manuscript is now with our production department. 

Kind regards, 

on behalf of

Dr. Beatriz Talavera-Velasco 

Academic Editor

PLOS ONE